# Palladium-Catalyzed Direct (Het)arylation Reactions of Benzo[1,2-d:4,5-d′]bis([1,2,3]thiadiazole and 4,8-Dibromobenzo[1,2-d:4,5-d′]bis([1,2,3]thiadiazole)

**DOI:** 10.3390/molecules28093977

**Published:** 2023-05-08

**Authors:** Timofey N. Chmovzh, Timofey A. Kudryashev, Daria A. Alekhina, Oleg A. Rakitin

**Affiliations:** 1N. D. Zelinsky Institute of Organic Chemistry, Russian Academy of Sciences, 119991 Moscow, Russia; tim1661@yandex.ru (T.N.C.); tp12345678@yandex.ru (T.A.K.); alekhinadariaalex@gmail.com (D.A.A.); 2Nanotechnology Education and Research Center, South Ural State University, 454080 Chelyabinsk, Russia; 3Department of Chemistry, Moscow State University, 119899 Moscow, Russia; 4Higher Chemical College, Mendeleev University of Chemical Technology of Russia, 125047 Moscow, Russia

**Keywords:** sulfur-nitrogen heterocycles, benzo[1,2-d:4,5-d′]bis([1,2,3]thiadiazole), 4,8-dibromobenzo[1,2-d:4,5-d′]bis([1,2,3]thiadiazole), direct (het)arylation palladium catalyzed reactions

## Abstract

Palladium-catalyzed direct (het)arylation reactions of strongly electron-withdrawing tricyclic benzo[1,2-d:4,5-d′]bis([1,2,3]thiadiazole) and its 4,8-dibromo derivative were studied; the conditions for the selective formation of mono- and bis-aryl derivatives were found. The reaction of 4,8-dibromobenzo[1,2-d:4,5-d′]bis([1,2,3]thiadiazole) with thiophenes in the presence of palladium acetate as a catalyst and potassium pivalate as a base, depending on the conditions used, selectively gave both mono- and bis-thienylated benzo-bis-thiadiazoles in low to moderate yields; arenes were found to be inactive in these reactions. It was discovered that direct C–H arylation of benzo[1,2-d:4,5-d′]bis([1,2,3]thiadiazole with bromo(iodo)arenes and -thiophenes in the presence of Pd(OAc)2 and di-tert-butyl(methyl)phosphonium tetrafluoroborate salt is a powerful tool for the selective formation of 4-mono- and 4,8-di(het)arylated benzo-bis-thiadiazoles. Oxidative double C–H hetarylation of benzo[1,2-d:4,5-d′]bis([1,2,3]thiadiazole with thiophenes in the presence of Pd(OAc)_2_ and silver (I) oxide in DMSO was successfully employed to prepare bis-thienylbenzo-bis-thiadiazoles in moderate yields.

## 1. Introduction

π-Conjugated organic molecules have attracted much attention in optoelectronic devices due to their ability to optimize many physical properties, such as light absorption, light emission, charge carrier mobility, conductivity, and others [1]. Various combinations of electron-donating (D) and electron-withdrawing (A) groups, linked either directly or preferably through π-conjugated bridges (π), have been used in organic chromophores to tune band gap levels and optoelectronic properties. The selection of donor and acceptor fragments is fundamentally important for achieving the best characteristics of organic dyes. An essential role of electron-deficient π-conjugated building blocks is to reduce the band gap by promoting intramolecular charge transfer (ICT) [2,3]. Although a number of heterocyclic acceptors have been extensively studied [4], 2,1,3-benzothiadiazole and its 4,7-disubstituted derivatives are the most promising acceptor units due to their strong electron-withdrawing properties, intense light absorption, and excellent photochemical stability [5,6]. Nevertheless, attempts have been made to increase the electron-withdrawing strength of the benzothiadiazole moiety by introducing fluorine atoms into positions 5 and 6 of the benzene ring [7], replacing the benzene ring with a pyridazine ring [8,9], and heteroannelation in positions 5 and 6 with another thiadiazole ring to form a strong acceptor building block, such as benzo[1,2-*c*:4,5-*c*′]bis[1,2,5]thiadiazole (**BBT**) with the lowest LUMO energy (Figure 1) [10]. The BBT isomer, benzo[1,2-*d*:4,5-*d*′]bis([1,2,3]thiadiazole) (**isoBBT**), has recently been found to have promising electron-accepting properties [11]. It was shown that 4,8-dibromobenzo[1,2-*d*:4,5-*d*′]bis([1,2,3]thiadiazole) can successfully participate in palladium-catalyzed Suzuki–Miyaura and Stille cross-coupling reactions with selective formation of mono- and bis-arylated heterocycles, which can be considered as useful building blocks for DSSC and OLED components [12].

Although traditional methods of C–C bond formation have proved to be effective for **isoBBT** derivatives [12], modern environmental safety requirements require a reduction in the number of technological stages, as well as the abandonment of the use of toxic (organotin) and flammable (butyllithium) reagents in these reactions. One way to eliminate these shortcomings is palladium-activated direct (het)arylation by the reaction of some (het)aryl derivatives with others [13]. With the help of these efficient synthetic tools, many π-conjugated molecules have been obtained [14,15,16]. There are three approaches to such a transformations of **isoBBT** derivatives: 1. reaction of halogen **iso-BBT** derivatives (i.e., 4,8-dibromobenzo[1,2-*d*:4,5-d′]bis([1,2,3]thiadiazole)) with C–H (het)aryls; 2. reaction of benzo[1,2-*d*:4,5-*d*′]bis([1,2,3]thiadiazole) **1** with halogen (het)aryls; and 3. double oxidative arylation of benzo[1,2-*d*:4,5-d′]bis([1,2,3]thiadiazole) with C-H (het)aryl. These three routes were investigated for 2,1,3-benzothiadiazole (**BTD**) derivatives (Figure 1). The reaction of 4,7-dibromobenzo[c][1,2,5]thiadiazole with arenes and hetarenes (Figure 1, path 1) is most often carried out in the presence palladium (II) acetate as a palladium catalyst and potassium acetate [17,18,19] or potassium pivalate [20,21,22,23,24] as bases in N,N-dimethylacetamide (DMA). In some cases, triphenyl- [25] or tricyclohexylphosphine [26] have been used as ligands. A combination of reagents, including Pd(OAc)_2_, tricyclohexylphosphine tetrafluroborate salt (Cy_3_PHBF_4_), sodium *tert*-butoxide and neodecanoic acid was effective for the synthesis of 4,7-bis(5-hexyl-2-thienyl)benzo[c][1,2,5]thiadiazole [27]. tris-(Dibenzylideneacetone)dipalladium(0) (Pd_2_(dba)_3_) together with potassium pivalate as a base and tris(*o*-methoxyphenyl)phosphine as a ligand was successfully employed for arylation of 4,7-dibromobenzo[c][1,2,5]thiadiazole [28,29,30,31]. Thiazolyl derivatives of benzo[c][1,2,5]thiadiazole were prepared in good yields using palladacycle Herrmann complex (trans-di(μ-acetato)-bis[*o*-(di-*o*-tolylphosphino)-benzyl]dipalladium(II)), cesium pivalate as base and tris(*o*-methoxyphenyl)phosphine as ligand [32].

Unsubstituted benzo[c][1,2,5]thiadiazole reacted with bromoarenes or hetarenes (Figure 1, path 2) by catalysis of palladium (II) acetate in the presence of potassium pivalate in DMA at a high temperature 150 °C with successful formation of mono- and bis-(het)aryl derivatives [33]. The use of di-*tert*-butyl(methyl)phosphonium tetrafluoroborate salt (PBu^t^_2_Me·HBF_4_) in toluene made it possible to lower the reaction temperature to 120 °C and extend the reaction scope for 5-mono- and 5,6-difluoro(cyano)benzo[c][1,2,5]thiadiazoles [34,35].

Selective Pd-catalyzed (Pd(OAc)_2_) thienylation of benzo[c][1,2,5]thiadiazoles with thiophenes (Figure 1, path 3) in DMSO via double oxidative C–H functionalization was discovered in 2014 by the Zhang group [36,37]. The reaction proceeds under mild reaction conditions, providing a series of unsymmetrical and symmetrical **BTD**–thiophenes with high efficiency and excellent functional group compatibility. Silver oxide acted as an oxidizing agent; in some cases, Pd(OTf)_2_ gave higher yields of dithienylated benzo[c][1,2,5]thiadiazoles [37].

There is only one example of direct C–H hetarylation of tricyclic benzo-bis-thiadiazoles: the synthesis of 4,8-bis(5-(triisopropylsilyl)thiophen-2-yl)benzo[1,2-*d*:4,5-*d*′]bis([1,2,3]thiadiazole) in low yield upon treatment of **isoBBT 1** with palladium (II) acetate in the presence of potassium pivalate and di-*tert*-butyl(methyl)phosphonium tetrafluoroborate salt (P^t^Bu_2_Me·HBF_4_) in toluene at 120 °C (Figure 2) [11].

To elucidate the applicability of direct C–H (het)arylation reactions of tricyclic benzo-bis-thiadiazoles, this paper describes the study of the reaction of benzo[1,2-*d*:4,5-*d*′]bis([1,2,3]thiadiazole) **1** and its 4,8-dibromo derivative **2** with aromatic and heterocyclic compounds.

## 2. Results and Discussion

### 2.1. Palladium-Catalyzed (Het)arylation Reactions of 4,8-Dibromobenzo[1,2-d:4,5-d′]bis([1,2,3]thiadiazole) 2

The optimal conditions for the selective synthesis of mono-**4** and bis-**5** coupling products were calculated for the reaction of 4,8-dibromobenzo[1,2-*d*:4,5-*d*′]bis([1,2,3]thiadiazole) **2** with (2-ethylhexyl)thiophene **3a** in the presence of various palladium catalysts and organic ligands. The results of this study are summarized in Table 1. It was found that by using Pd(OAc)_2_ with potassium pivalate as a base in toluene, both mono-**4a**- and bis-aryl derivatives **5a** can be obtained. The nature of the solvent and ligand, the temperature of the chemical transformation, and the excess of the reagent significantly affected the results of the reactions (Table 1). Unexpectedly, carrying out the reaction in the frequently used solvent DMA [17,18,19,20] resulted only in the decomposition of the starting dibromide **2** (Table 1, entry 1). Refluxing in the aromatic solvent, toluene, led to the disappearance of the starting bicycle **2** with the formation of the target product **4a** in moderate yield (Table 1, entry 2). An increase in the reaction temperature to 130 °C and an increase in the amount of the starting thiophene to two equivalents gave bis-coupling product **5a** in a yield close to that of mono-product **4a** (Table 1, entry 3). An unexpected fact was that the use of ligands such as tri-*tert*-butylphosphine (Bu^t^_3_P), bis(diphenylphosphino)ferrocene (dppf) or XPhos, PBu^t^_2_MeHBF_4_, both in toluene and in DMA, stopped the formation of products **4a** and **5a**; in these cases, the starting heterocycle **2** decomposed slowly under the reaction conditions (Table 1, entries 4–9). The optimal conditions were extended to other thiophene derivatives **3b-d**; mono- and bis-dithienylated derivatives were isolated in moderate yields (Table 1, entries 12–17). Attempts to carry out the C–H arylation reaction involving aromatic compounds such as toluene or xylene using various catalytic systems were not successful; starting dibromide **2** was isolated in high yields. Thus, we have shown that the C–H arylation reactions of dibromide **2** proceeded only with heteroaromatic thiophene derivatives **3a-d** and selectively led to the formation of mono- and bis-thienyl derivatives in moderate yields.

### 2.2. Palladium-Catalyzed (Het)arylation Reactions of Benzo[1,2-d:4,5-d′]bis([1,2,3]thiadiazole) **1**

Palladium-catalyzed direct arylation reactions of non-halogenated aromatic electron-withdrawing heterocycles are much less studied. The results of the reaction of tricycle 1 with 2-bromo-5-(2-ethylhexyl)thiophene **6a(Br)** as a halogen-containing substrate are summarized in Table 2. Refluxing in toluene in the presence of palladium acetate (Pd(OAc)_2_) and potassium pivalate (PivOK) resulted in partial decomposition of the starting bicycle **1** without the formation of target products **7a** and **5a** (Table 2, entry 1). The introduction of such ligands as tri-*tert*-butylphosphine (But_3_P) or bis(diphenylphosphino)ferrocene (dppf) did not activate the cross-coupling reaction (Table 2, entries 3,4), but the employing of XPhos led to the formation of a monocoupling product **7a** with a low yield (Table 2, entry 2). The use of such palladium catalysts as tetrakis(triphenylphosphine)palladium (Pd(PPh_3_)_4_), tris(dibenzylideneacetone)dipalladium (Pd_2_(dba)_3_), and bis(triphenylphosphine)palladium chloride (PdCl_2_(PPh_3_)_2_) also did not run the cross-coupling reaction (Table 2, entries 5,6,8). The best results were shown by a catalytic system based on (Pd(OAc)_2_) and di-*tert*-butyl(methyl)phosphonium tetrafluoroborate salt (P(Bu^t^)_2_MeHBF_4_) [36]. If the reaction of benzo[1,2-*d*:4,5-*d*′]bis([1,2,3]thiadiazole) **1** was carried out in refluxing toluene in the presence of potassium pivalate, then the bis-coupling product **7a** was formed (Table 2, entry 9). Long-term reflux in toluene in the presence of Pd(OAc)_2_ and P(Bu^t^)_2_MeHBF_4_ led to the formation of compound **7a** in 45% yield (Table 2, entry 10). It was shown that the replacement of toluene by higher boiling xylene (130 °C) shifted the C-H arylation reaction towards the bis-coupling product **5a** in a good yield of 55% (Table 2, entry 11). The use of DMA or DMF as a solvent did not lead to the formation of cross-coupling products (Table 2, entries 12,13). Treatment of tricyle **1** with one equivalent of 2-iodo-5-(2-ethylhexyl)thiophene in the presence of Pd(OAc)_2_) and P(Bu^t^)_2_MeHBF_4_) led to the formation of a mixture of mono-**7a** and bis-**5a** substituted products in a ratio of 2:1 (Table 2, entry 14). Increasing the amount of iodine derivative **6a(I)** to two equivalents and replacing toluene with xylene resulted in the selective formation of the bis-coupling product **5a** in 54% yield (Table 1, entry 15). 2-Chloro-5-(2-ethylhexyl)thiophene gave under these conditions the mono-coupling product **7a** in trace amounts of 2% (Table 2, entry 16).

The optimal conditions for the cross-coupling reaction (Pd(OAc)_2_ and PBu^t^_2_MeHBF_4_ catalytic system in refluxing toluene at 110 °C or in xylene at 130 °C) were extended to halogenated derivatives of thiophene and benzene **6b-j**. If for 2-bromothiophenes **6a-c,e(Br)** the hetarylation reactions proceeded selectively and with moderate yields of both mono-**7** and bis-**5** products, then for bromoarenes the chemical transformation led to a lower yield of mono- and bis-coupling products (Table 3, entries 9,10). The replacement of bromobenzene **6f(Br)** by the more reactive iodobenzene **6f(I)** made it possible to significantly increase the yield of both mono-coupling **7f** and bis-coupling **5f** products (Table 3, entries 11,12). It was shown that the use of iodobenzenes **6(I)** in the reaction with tricycle **1** gave the target products **7** and **5** in moderate yields (Table 3, entries 13–20).

### 2.3. Palladium-Catalyzed Oxidative (Het)arylation Reactions of Benzo[1,2-d:4,5-d′]bis([1,2,3]thiadiazole) **1**

Oxidative hetarylation reactions of tricycle **1** with thiophene derivatives were studied using (2-ethylhexyl)thiophene **3a**, palladium trifluoroacetate and acetate as catalysts under the action of silver (I) oxide (Ag_2_O) as an oxidizing agent in dimethyl sulfoxide as described for BTD derivatives (see Figure 1, path 3). Surprisingly, palladium trifluoroacetate did not catalyze this hetarylation reaction (Table 4, entry 1). The use of palladium acetate instead of palladium trifluoroacetate led to the formation of a mixture of mono-**7a** and bis-**5a** coupling products (Table 4, entry 2). We investigated the possibility of replacing silver oxide with silver salts such as silver acetate (AgOAc), silver nitrate (AgNO_3_), silver tetrafluoroborate (AgBF_4_), and silver perchlorate (AgClO_4_). It was shown that in the case of silver acetate, the total yield of the mixture of products **7a** and **5a** was only 25%, while in the case of silver nitrate, compound **5a** was isolated in 4% yield, and the use of silver tetrafluoroborate and silver perchlorate did not lead to the formation of thienylated products (Table 4, entries 3–6). Reducing the amount of thiophene derivative **3a** to one equivalent also gave a mixture of mono- and bis-derivatives in low yield with a significant predominance of mono-derivative **7a** (Table 4, entry 7) and using three equivalents of **3a**, together with increasing the reaction time to 48 h, gave the highest yield of bis-product **5a**, 55% (Table 4, entry 8). These conditions were extended to other thiophene derivatives **3b,c,e,** to produce bis-coupling products **5** in moderate to low yields (Table 4, entries 10–12). Attempts to carry out the reaction of oxidative arylation with benzene and toluene were unsuccessful; as a result, only a gradual decomposition of the starting tricycle **1** was observed.

### 2.4. Comparison of Suzuki and Stille Cross-Coupling Reactions with Direct (Het)arylation Reactions of Benzo[1,2-d:4,5-d′]bis([1,2,3]thiadiazole) **1** and 4,8-dibromo Derivative **2**

In order to compare the results of direct (het)arylation reactions of benzo[1,2-*d*:4,5-*d*′]bis([1,2,3]thiadiazole) **1** and its dibromo derivative **2** with classical cross-coupling reactions, we analyzed the results obtained in this work using data on the Suzuki and Stille reactions of dibromo derivative **2** described in [12]. The data are summarized in Figure 3.

We recently found that Stille coupling of 4,8-dibromobenzo[1,2-*d*:4,5-*d*′]bis([1,2,3]thiadiazole) **2** gave good yields of bis-arylated heterocycles **4** (55–73%, path 3), and the Suzuki–Miyaura reaction led to the selective formation of both mono- **4** (60–72%, path 1) and bis-(het)arylated **5** (50–67%, path 2) benzo[1,2-*d*:4,5-*d*′]bis([1,2,3]thiadiazoles) [12]. In this paper, we have shown that direct arylation of dibromotricycle **2** is successful only for thiophene derivatives and afforded approximately two times lower yields of mono-**4Th** (31–43%, path 5) and bis-**5Th** products (29–40%, path 6); arenes did not react with tricycle **2** at all. Even if we take into account that the yields of boronic esters and tributylstannyl thiophene derivatives from unsubstituted thiophenes are known to be below 100%, it seems that this direct arylation variant (paths 5 and 6) cannot compete with the Suzuki and Stille reactions for compound **2** (paths 1–4).

Two other variants of the direct (het)arylation reaction turned out to be more useful for preparation of (het)arylbenzo-bis-thiadiazoles. Thus, path 7, the direct arylation reaction of benzo-bis-thiadiazole **1** with halogenated thiophenes and arenes, makes it possible to obtain mono-derivatives **7**, which are inaccessible by other methods. Despite the fact that the yields of bis-aryl derivatives **5** in path 8 are somewhat lower (20–55%) than in the Suzuki and Stille reactions (paths 2 and 4), one should take into account the fact that dibromotricycle **2** is obtained from unsubstituted tricycle **1** with a yield of 40% [12], which practically equalizes the yields in the preparation of compounds **5** from unsubstituted tricycle **1** by its bromination followed by Suzuki and Stille reactions (paths 2 and 4) and direct (het)arylation with bromo(iodo)arenes and thiophenes (path 8). When comparing these methods, it should be taken into account that in direct (het)arylation there is no need to obtain boronic esters and trialkylstannyl derivatives, which usually require the use of flammable butyllithium and harmful tin compounds.

Oxidative hetarylation of compound **1** may be of particular interest for the preparation of bis-hetaryl derivatives **5Th**. Readily accessible heterocycle **1** and often commercially available thiophenes are involved in the reaction, which makes it possible to significantly reduce the number of steps in the synthesis of bis-thienylated benzo-bis-thiadiazoles **5Th** practically without reducing their yields. An important advantage of the last two variants of direct hetarylation (paths 8 and 9) is the selectivity of these processes, which greatly simplifies the procedure for isolating the final compounds. We found that refluxing dibromide **2** and tricycle **1** in toluene for 24 h resulted in their partial decomposition to a mixture of unidentifiable compounds, which, in turn, may also be the cause of low or moderate yields of C–H arylation reaction products.

## 3. Experimental Section

### 3.1. Materials and Reagents

The chemicals were purchased from commercial sources (Sigma-Aldrich, St. Louis, MO, USA) and used as received. Benzo[1,2-*d*:4,5-*d*′]bis([1,2,3]thiadiazole) **1** [38], 4,8-dibromobenzo[1,2-*d*:4,5-*d*′]bis([1,2,3]thiadiazole) **2** [12], 2-(2-ethylhexyl)thiophene **6a** [39], 2,2′-bithiophene **6d** [40], and [2,2′-bithiophen]-5-yltrimethylsilane **6e** [41] were prepared according to the published methods and characterized by NMR spectra. All synthetic operations were performed under a dry argon atmosphere. Toluene and xylene were distilled over Na. DMSO was distilled over CaH_2_.

### 3.2. Analytical Instruments

The melting points were determined on a Kofler hot-stage apparatus and were uncorrected. ^1^H and ^13^C NMR spectra were taken with a Bruker AM-300 machine (Bruker Ltd., Moscow, Russia) with TMS as the standard. *J* values are given in Hz. MS spectra (EI, 70 eV) were obtained with a Finnigan MAT INCOS 50 instrument (Thermo Finnigan LLC, San Jose, CA, USA). High-resolution MS spectra were measured on a Bruker micrOTOF II instrument (Bruker Ltd., Moscow, Russia) using electrospray ionization (ESI). IR spectra were measured with a Bruker “Alpha-T” instrument (Bruker, Billerica, MA, USA) in KBr pellets, details at the Appendix A.

### 3.3. General Procedure for the Synthesis of Mono-Substituted Products **4** from 4,8-Dibromobenzo[1,2-d:4,5-d′]bis([1,2,3]thiadiazole) **2** (Procedure A)

Pd(OAc)_2_ (9 mg, 0.042 mmol), pivalic acid (28 mg, 0.28 mmol) and K_2_CO_3_ (38 mg, 0.28 mmol) were added to a solution of 4,8-dibromobenzo[1,2-*d*:4,5-*d*′]bis([1,2,3]thiadiazole) **2** (100 mg, 0.28 mmol), thiophene **3a-d** (0.28 mmol) in anhydrous toluene (8 mL). The resulting mixture was degassed by argon in a sealed vial. The resulting mixture was then stirred at 110 °C for the time shown in Table 1. On completion (monitored by TLC), the mixture was poured into water and extracted with CH_2_Cl_2_ (3 × 35 mL). The combined organic layers were washed with brine, dried over MgSO_4_, filtered, and concentrated under reduced pressure. The crude product was purified by column chromatography.

### 3.4. General Procedure for the Synthesis of Bis-Substituted Products **5** from 4,8-Dibromobenzo[1,2-d:4,5-d′]bis([1,2,3]thiadiazole) **2** (Procedure B)

Pd(OAc)_2_ (9 mg, 0.042 mmol), pivalic acid (56 mg, 0.56 mmol) and K_2_CO_3_ (76 mg, 0.56 mmol) were added to a solution of 4,8-dibromobenzo[1,2-*d*:4,5-*d*′]bis([1,2,3]thiadiazole) **2** (100 mg, 0.28 mmol), thiophene **3a-d** (0.56 mmol) in anhydrous xylene (8 mL). The resulting mixture was degassed by argon in a sealed vial. The resulting mixture was then stirred at 130 °C for the time shown in Table 1. On completion (monitored by TLC), the mixture was poured into water and extracted with CH_2_Cl_2_ (3 × 35 mL). The combined organic layers were washed with brine, dried over MgSO_4_, filtered, and concentrated under reduced pressure. The crude product was purified by column chromatography.

### 3.5. General Procedure for the Preparation of Mono-Substituted Products **7** from Benzo[1,2-d:4,5-d′]bis([1,2,3]thiadiazole) **1** (Procedure C)

Pd(OAc)_2_ (9 mg, 0.042 mmol), (P(Bu^t^)_2_MeHBF_4_) (19 mg, 0.18 mmol), pivalic acid (105 mg, 1.03 mmol) and K_2_CO_3_ (142 mg, 1.03 mmol) were added to a solution of benzo[1,2-*d*:4,5-*d*′]bis([1,2,3]thiadiazole) **1** (200 mg, 1.03 mmol), bromide or iodide **6a-d,f-j(X)** (1.03 mmol) in anhydrous toluene (8 mL). The resulting mixture was degassed by argon in a sealed vial. The resulting yellow mixture was then stirred at 110 °C for the time shown in Table 3. On completion (monitored by TLC), the mixture was poured into water and extracted with CH_2_Cl_2_ (3 × 35 mL). The combined organic layers were washed with brine, dried over MgSO_4_, filtered, and concentrated under reduced pressure. The crude product was purified by column chromatography.

### 3.6. General Procedure for the Preparation of Bis-Substituted Products **5** from Benzo[1,2-d:4,5-d′]bis([1,2,3]thiadiazole) **1** (Procedure D)

Pd(OAc)_2_ (9 mg, 0.042 mmol), (P(Bu^t^)_2_MeHBF_4_) (19 mg, 0.18 mmol), pivalic acid (210 mg, 2.06 mmol) and K_2_CO_3_ (284 mg, 2.06 mmol) were added to a solution of benzo[1,2-*d*:4,5-*d*′]bis([1,2,3]thiadiazole) **1** (200 mg, 1.03 mmol), bromide or iodide **6a-d,f-j(X)** (2.06 mmol) in anhydrous xylene (8 mL). The resulting mixture was stirred and degassed by argon in a sealed vial. The resulting yellow mixture was then stirred at 130 °C for the time shown in Table 3. On completion (monitored by TLC), the mixture was poured into water and extracted with CH_2_Cl_2_ (3 × 35 mL). The combined organic layers were washed with brine, dried over MgSO_4_, filtered, and concentrated under reduced pressure. The crude product was purified by column chromatography.

### 3.7. General Procedure for the Preparation of Bis-Substituted Products **5** under C-H Oxidative Coupling Conditions (Procedure E)

Ag_2_O (234 mg, 1.02 mmol) and Pd(OAc)_2_ (9 mg, 0.042 mmol) were added to a solution of benzo[1,2-*d*:4,5-*d*′]bis([1,2,3]thiadiazole) **1** (100 mg, 0.51 mmol) and thiophene **3a-c,e** (1.53 mmol) in dry DMSO (5 mL). The resulting mixture was degassed by argon in a sealed vial. The resulting yellow mixture was then stirred at 90 °C for the time shown in Table 4. On completion (monitored by TLC), the mixture was poured into water and extracted with CH_2_Cl_2_ (3 × 35 mL). The combined organic layers were washed with brine, dried over MgSO_4_, filtered, and concentrated under reduced pressure. The crude product was purified by column chromatography.

### 3.8. Preparation of Preparation of 4-(5-(2-Ethylhexyl)thiophen-2-yl)benzo[1,2-d:4,5-d′]bis([1,2,3]thiadiazole) **7a** under C-H Oxidative Coupling Conditions (Procedure F)

Ag_2_O (117 mg, 0.51 mmol) and Pd(OAc)_2_ (9 mg, 0.042 mmol) were added to a solution of benzo[1,2-*d*:4,5-*d*′]bis([1,2,3]thiadiazole) **1** (100 mg, 0.51 mmol) and 2-(2-ethylhexyl)thiophene **3a** (0.51 mmol) in dry DMSO (5 mL). The resulting mixture was degassed by argon in a sealed vial. The resulting yellow mixture was then stirred at 90 °C for the time shown in Table 4. On completion (monitored by TLC), the mixture was poured into water and extracted with CH_2_Cl_2_ (3 × 35 mL). The combined organic layers were washed with brine, dried over MgSO_4_, filtered, and concentrated under reduced pressure. The crude product was purified by column chromatography.

4-Bromo-8-(5-(2-ethylhexyl)thiophen-2-yl)benzo[1,2-*d*:4,5-*d*′]bis([1,2,3]thiadiazole) (**4a**).

Yellow solid, 56 mg (43%) (procedure A), eluent-CH_2_Cl_2_:hexane, 1:1 (*v*/*v*). R_f_ = 0.6 (CH_2_Cl_2_). Mp = 57–60 °C. (lit. mp 57–60 °C [12]). The data of the ^1^H and ^13^C NMR spectra correspond to the literature data [12]. ^1^H NMR (300 MHz, CDCl_3_): *δ* 8.06 (d, *J* = 3.8, 1H), 7.01 (d, *J* = 3.8, 1H), 2.90 (d, *J* = 6.8, 2H), 1.77–1.69 (m, 1H), 1.45–1.31 (m, 8H), 0.97–0.89 (m, 6H).

4-Bromo-8-(thiophen-2-yl)benzo[1,2-*d*:4,5-*d*′]bis([1,2,3]thiadiazole) (**4b**).

Yellow solid, 32 mg (33%) (procedure A), eluent-CH_2_Cl_2_:hexane, 1:1 (*v*/*v*).R_f_ = 0.4 (CH_2_Cl_2_). Mp = 198–200 °C. (lit. mp 198–200 °C [12]). The data of the ^1^H and ^13^C NMR spectra correspond to the literature data [12]. ^1^H NMR (300 MHz, CDCl_3_): *δ* 8.20 (d, *J* = 3.9, 1H), 7.76 (d, *J* = 5.2, 1H), 7.36 (t, *J* = 4.5, 1H).

4-Bromo-8-(4-hexylthiophen-2-yl)benzo[1,2-*d*:4,5-*d*′]bis([1,2,3]thiadiazole) (**4c**).

Yellow solid, 43 mg (35%) (procedure A), eluent-CH_2_Cl_2_:hexane, 1:2 (*v*/*v*). R_f_ = 0.6 (CH_2_Cl_2_:hexane, 1:1 (*v*/*v*)). Mp = 67–69 °C. (lit. mp 67–69 °C [12]). The data of the ^1^H and ^13^C NMR spectra correspond to the literature data [12]. ^1^H NMR (300 MHz, CDCl_3_): *δ* 8.09 (s, 1H), 7.35 (s, 1H), 2.76 (t, *J* = 7.7, 2H), 1.73 (p, *J* = 7.2, 2H), 1.42–1.31 (m, 6H), 0.91 (t, *J* = 6.9, 3H).

4-([2,2′-Bithiophen]-5-yl)-8-bromobenzo[1,2-*d*:4,5-*d*′]bis([1,2,3]thiadiazole) (**4d**).

Red solid, 44 mg (31%) (procedure A), eluent-CH_2_Cl_2_:hexane, 1:1 (*v*/*v*). R_f_ = 0.4 (CH_2_Cl_2_:hexane, 1:1 (*v*/*v*)). Mp = 130–132 °C. (lit. mp 130–132 °C [12]). The data of the ^1^H and ^13^C NMR spectra correspond to the literature data [12]. ^1^H NMR (300 MHz, CDCl_3_): *δ* 8.09 (d, *J* = 4.0, 1H), 7.40–7.34 (m, 3H), 7.14–7.07 (m, 1H).

4,8-Bis(5-(2-ethylhexyl)thiophen-2-yl)benzo[1,2-*d*:4,5-*d*′]bis([1,2,3]thiadiazole) (**5a**).

Red solid, 65 mg (40%, procedure B), or 329 mg (55%, procedure D), or 159 mg (55%, procedure E), eluent-CH_2_Cl_2_:hexane, 1:4 (*v*/*v*). R_f_ = 0.7 (CH_2_Cl_2_:hexane, 1:4 (*v*/*v*)). Mp = 78–80 °C. (lit. mp 78–80 °C [12]). The data of the ^1^H and ^13^C NMR spectra correspond to the literature data [12]. ^1^H NMR (300 MHz, CDCl_3_): *δ* 8.01 (d, *J* = 3.8, 2H), 7.00 (d, *J* = 3.8, 2H), 2.90 (d, *J* = 6.8, 4H), 1.73 (p, *J* = 5.9, 2H), 1.46–1.30 (m, 16H), 0.94–0.90 (m, 12H).

4,8-Di(thiophen-2-yl)benzo[1,2-*d*:4,5-*d*′]bis([1,2,3]thiadiazole) (**5b**).

Red solid, 36 mg (36%, procedure B), or 129 mg (35%, procedure D), or 64 mg (36%, procedure E), eluent-CH_2_Cl_2_:hexane, 1:2 (*v*/*v*). R_f_ = 0.5 (CH_2_Cl_2_:hexane, 1:1 (*v*/*v*)). Mp > 250 °C (lit. mp > 250 °C [12]). The data of the ^1^H and ^13^C NMR spectra correspond to the literature data [12]). ^1^H NMR (300 MHz, CDCl_3_): *δ* 8.20 (dd, *J* = 3.8, 1.2, 2H), 7.74 (dd, *J* = 5.1, 1.2, 2H), 7.37 (dd, *J* = 5.1, 3.8, 2H).

4,8-Bis(4-hexylthiophen-2-yl)benzo[1,2-*d*:4,5-*d*′]bis([1,2,3]thiadiazole) (**5c**).

Red solid, 42 mg (29%, procedure B), 184 mg (34%, procedure D) or 102 mg (39%, procedure E), eluent-CH_2_Cl_2_:hexane, 1:3 (*v*/*v*). R_f_ = 0.7 (CH_2_Cl_2_:hexane, 1:1 (*v*/*v*)). Mp = 134–136 °C. (lit. mp 134–136 °C [12]). The data of the ^1^H and ^13^C NMR spectra correspond to the literature data [12]). ^1^H NMR (300 MHz, CDCl_3_): *δ* 8.09 (d, *J* = 1.3, 2H), 7.32 (d, *J* = 1.3, 2H), 2.77 (t, *J* = 7.7, 4H), 1.75 (p, *J* = 7.6, 4H), 1.45–1.33 (m, 12H), 0.94–0.89 (m, 6H).

4,8-Di([2,2′-bithiophen]-5-yl)benzo[1,2-*d*:4,5-*d*′]bis([1,2,3]thiadiazole) (**5d**).

Violet solid, 43 mg (30%, procedure B), eluent-CH_2_Cl_2_:hexane, 1:2 (*v*/*v*). R_f_ = 0.5 (CH_2_Cl_2_:hexane, 1:1 (*v*/*v*)). Mp > 250 °C. (lit. mp > 250 °C [12]). The data of the ^1^H and ^13^C NMR spectra correspond to the literature data [12]). ^1^H NMR (300 MHz, CDCl_3_): *δ* 8.07 (d, *J* = 4.1 Hz, 2H), 7.39 (d, *J* = 4.1 Hz, 3H), 7.37 (d, *J* = 5.1 Hz, 2H), 7.12–7.09 (m, 3H).

4,8-Bis(5′-(trimethylsilyl)-[2,2′-bithiophen]-5-yl)benzo[1,2-*d*:4,5-*d*′]bis([1,2,3]thiadiazole) (**5e**).

Violet solid, 266 mg (39%, procedure D), 126 mg (38%, procedure E), eluent-CH_2_Cl_2_/hexane, 1:2 (*v*/*v*). R_f_ = 0.1 (CH_2_Cl_2_:hexane, 1:1, (*v*/*v*)). Mp = 87–89 °C. IR ν_max_ (KBr, cm^−1^): 2961, 2924, 2853, 1727, 1497, 1453, 1400, 1370, 1317, 1289, 1261, 1098, 1023, 992, 800, 752, 694, 476. ^1^H NMR (300 MHz, CDCl_3_): δ 8.10 (d, *J* = 4.0, 2H), 7.45 (d, *J* = 3.4, 2H), 7.41 (d, *J* = 4.0, 2H), 7.22 (d, *J* = 3.5, 2H), 0.38 (s, 18H). ^13^C NMR (100 MHz, CDCl_3_): δ 154.0, 142.9, 142.0, 141.3, 139.0, 136.3, 135.1, 132.2, 126.4, 124.9, 120.4, −0.03 (TMS). MS (EI, 70eV), *m*/*z* (I, %): 700 ([M + 3]^+^, 4), 669 ([M + 2]^+^, 10), 668 ([M + 1]^+^, 25), 667 ([M]^+^, 45), 666 ([M − 1]^+^, 100), 610 (25), 595 (6), 534 (15), 519 (3), 505 (6), 43 (3).

4,8-Diphenylbenzo[1,2-*d*:4,5-*d*′]bis([1,2,3]thiadiazole) (**5f**).

Yellow solid, 178 mg (50%, procedure D), eluent-CH_2_Cl_2_:hexane, 1:2 (*v*/*v*). R_f_ = 0.5 (CH_2_Cl_2_:hexane, 1:1 (*v*/*v*)). Mp > 250 °C. (lit. mp > 250 °C [12]). The data of the ^1^H and ^13^C NMR spectra correspond to the literature data [12]). ^1^H NMR (300 MHz, CDCl_3_): *δ* 7 8.01 (d, *J* = 7.0, 4H), 7.69–7.58 (m, 6H).

4,8-Di-*p*-tolylbenzo[1,2-*d*:4,5-*d*′]bis([1,2,3]thiadiazole) (**5g**).

Yellow solid, 192 mg (50%, procedure D), eluent-CH_2_Cl_2_:hexane, 1:2 (*v*/*v*). R_f_ = 0.4 (CH_2_Cl_2_:hexane 1:1 (*v*/*v*)). Mp > 250 °C. The data of the ^1^H and ^13^C NMR spectra correspond to the literature data [12]). ^1^H NMR (300 MHz, CDCl_3_): *δ* 7.90 (d, *J* = 7.9, 4H), 7.90 (d, *J* = 7.9, 4H), 2.52 (s, 6H).

4,8-Bis(4-methoxyphenyl)benzo[1,2-*d*:4,5-*d*′]bis([1,2,3]thiadiazole) (**5h**).

Orange solid, 230 mg (55%, procedure D), eluent-CH_2_Cl_2_:hexane, 1:1 (*v*/*v*). R_f_ = 0.2 (CH_2_Cl_2_:hexane, 1:1 (*v*/*v*)). Mp > 250 °C. (lit. mp > 250 °C [12]). The data of the ^1^H and ^13^C NMR spectra correspond to the literature data [12]). ^1^H NMR (300 MHz, CDCl_3_): *δ* 7.97 (d, *J* = 8.3, 4H), 7.17 (d, *J* = 8.3, 4H), 3.95 (s, 6H).

Dimethyl 4,4′-(benzo[1,2-*d*:4,5-*d*′]bis([1,2,3]thiadiazole)-4,8-diyl)dibenzoate (**5i**).

Yellow solid, 233 mg (49%, procedure D), eluent–CH_2_Cl_2_/hexane, 1:2 (*v*/*v*). R_f_ = 0.1 (CH_2_Cl_2_:hexane, 1:1, (*ν*/*ν*)). Mp > 250 °C. IR ν_max_ (KBr, cm^−1^): 2954, 2925, 2854, 1724, 1642, 1608, 1430, 1413, 1317, 1287, 1209, 1189, 1112, 1012, 960, 860, 825, 766, 695, 646, 567. ^1^H NMR (300 MHz, CDCl_3_): δ 8.33 (d, *J* = 8.5, 4H), 8.10 (d, *J* = 8.5, 4H), 4.01 (s, 6H). ^13^C NMR (100 MHz, CDCl_3_): δ 166.1, 155.3, 141.7, 140.9, 131.7, 130.3, 129.6, 127.5, 52.2. HRMS (ESI-TOF), *m*/*z*: calcd for C_22_H_15_N_4_O_4_S_2_ [M + H]^+^, 463.0529, found, 463.0521. MS (EI, 70eV), *m*/*z* (I, %): 462 ([M]^+^, 10), 431 (4), 406 (12), 375 (11), 347 (60), 332 (4), 303 (7), 288 (25), 203 (8), 144 (40), 59 (100), 15 (25).

4,4′-(Benzo[1,2-*d*:4,5-*d*′]bis([1,2,3]thiadiazole)-4,8-diyl)bis(*N*,*N*-diphenylaniline) (**5j**).

Red solid, 175 mg (25%, procedure D), eluent-CH_2_Cl_2_:hexane, 1:2 (*v*/*v*). R_f_ = 0.5 (CH_2_Cl_2_:hexane, 1:1 (*v*/*v*)). Mp > 250 °C. (lit. mp > 250 °C [12]). The data of the ^1^H and ^13^C NMR spectra correspond to the literature data [12]). ^1^H NMR (300 MHz, CDCl_3_): *δ* 7.85 (d, *J* = 8.2, 4H), 7.35–7.30 (m, 8H), 7.24–7.08 (m, 16H).

4-(5-(2-Ethylhexyl)thiophen-2-yl)benzo[1,2-*d*:4,5-*d*′]bis([1,2,3]thiadiazole) (**7a**).

Orange solid, 179 mg (45%, procedure C), 69 mg (30%, procedure F), eluent-CH_2_Cl_2_/hexane, 1:2 (*v*/*v*). R_f_ = 0.4 (CH_2_Cl_2_:hexane, 1:1, (*ν*/*ν*)). Mp = 55–57 °C. IR ν_max_ (KBr, cm^−1^): 2958, 2923, 2855, 1618, 1507, 1457, 1389, 1324, 1282, 1262, 1144, 1078, 1032, 881, 861, 847, 812, 786, 739, 618, 547. ^1^H NMR (300 MHz, CDCl_3_): δ 9.10 (s, 1H), 8.09 (d, *J* = 3.7, 1H), 7.01 (d, *J* = 3.7, 1H), 2.91 (d, *J* = 6.8, 2H), 1.78–1.68 (m, 1H), 1.48–1.29 (m, 8H), 0.98–0.89 (m, 6H). ^13^C NMR (100 MHz, CDCl_3_): δ 158.3, 151.1, 140.8, 136.7, 135.1, 131.8, 127.0, 126.5, 123.6, 111.1, 41.6, 34.4, 32.5, 28.9, 25.7, 23.0, 14.1, 10.9. HRMS (ESI-TOF), *m*/*z*: calcd for C_18_H_21_N_4_S_3_ [M + H]^+^, 389.0923, found, 389.0921. MS (EI, 70eV), *m*/*z* (I, %): 390 ([M + 2]^+^, 3), 389 ([M + 1]^+^, 6), 388 ([M]^+^, 35), 360 (80), 332 (15), 261 (18), 248 (38), 233 (100), 69 (28), 57 (60), 41 (45), 29 (37).

4-(Thiophen-2-yl)benzo[1,2-*d*:4,5-*d*′]bis([1,2,3]thiadiazole) (**7b**).

Orange solid, 85 mg (30%, procedure C), or 50 mg (29%, procedure B), eluent–CH_2_Cl_2_/hexane, 1:2 (*v*/*v*). R_f_ = 0.3 (CH_2_Cl_2_:hexane, 1:1, (*v*/*v*)). Mp = 173–175 °C. IR ν_max_ (KBr, cm^−1^): 1636, 1532, 1437, 1432, 1393, 1328, 1286, 1258, 1142, 858, 812, 715, 666, 544. ^1^H NMR (300 MHz, CDCl_3_): δ 9.19 (s, 1H), 8.25 (d, *J* = 3.7, 1H), 7.75 (d, *J* = 5.0, 1H), 7.43–7.32 (m, 1H). ^13^C NMR (100 MHz, CDCl_3_): δ 158.4, 154.0, 140.9, 138.9, 137.6, 131.6, 130.5, 129.8, 128.6, 123.3, 112.1. HRMS (ESI-TOF), *m*/*z*: calcd for C_10_H_5_N_4_S_3_ [M + H]^+^, 276.9671, found, 276.9663. MS (EI, 70eV), *m*/*z* (I, %): 276 ([M]^+^, 6), 248 (75), 220 (10), 176 (11), 151 (100), 93 (25), 69 (95), 45 (12), 28 (5).

4-(4-Hexylthiophen-2-yl)benzo[1,2-*d*:4,5-*d*′]bis([1,2,3]thiadiazole) (**7c**).

Yellow solid, 140 mg (38%, procedure C), eluent-CH_2_Cl_2_/hexane, 1:2 (*v*/*v*). R_f_ = 0.4 (CH_2_Cl_2_:hexane, 1:1, (*v*/*v*)). Mp = 65–68 °C. IR ν_max_ (KBr, cm^−1^): 2956, 2924, 2853, 1640, 1540, 1513, 1494, 1451, 1398, 13754, 1333, 1287, 1249, 1188, 1081, 967, 854, 815, 775, 725, 661, 615, 522. ^1^H NMR (300 MHz, CDCl_3_): δ 9.16 (s, 1H), 8.14 (s, 1H), 7.34 (s, 1H), 3.18–2.61 (m, 2H), 1.79–1.70 (m, 2H), 1.40–1.30 (m, 6H), 0.91 (t, *J* = 8.0, 3H). ^13^C NMR (100 MHz, CDCl_3_): δ 157.2, 152.8, 144.1, 138.9, 136.2, 136.1, 132.2, 124.3, 122.5, 110.6, 30.7, 29.6, 29.5, 28.0, 21.6, 13.1 HRMS (ESI-TOF), *m*/*z*: calcd for C_16_H_17_N_4_S_3_ [M + H]^+^, 361.0610, found, 361.0606. MS (EI, 70eV), *m*/*z* (I, %): 362 ([M + 2]^+^, 3), 361 ([M + 1]^+^, 6), 360 ([M]^+^, 50), 332 (100), 248 (20), 235 (19), 220 (12), 165 (18), 120 (13), 69 (60), 43 (57), 29 (48).

4-(5′-(Trimethylsilyl)-[2,2′-bithiophen]-5-yl)benzo[1,2-*d*:4,5-*d*′]bis([1,2,3]thiadiazole) (**7e**).

Red solid, 140 mg (29%, procedure C), eluent-CH_2_Cl_2_/hexane, 1:2 (*v*/*v*). R_f_ = 0.3 (CH_2_Cl_2_:hexane, 1:1, (*v*/*v*)). Mp = 155–157 °C. IR ν_max_ (KBr, cm^−1^): 2958, 2924, 2853, 1724, 1641, 1494, 1464, 1364, 1279, 1263, 1187, 1081, 968, 892, 818, 725, 486. ^1^H NMR (300 MHz, CDCl_3_): δ 9.15 (s, 1H), 8.15 (d, *J* = 4.0, 1H), 7.44 (d, *J* = 3.5, 1H), 7.40 (d, *J* = 4.0, 1H), 7.22 (d, *J* = 3.5, 1H), 0.37 (s, 9H). ^13^C NMR (100 MHz, CDCl_3_): δ 158.6, 153.6, 143.1, 142.2, 141.2, 141.05, 136.8, 135.9, 135.2, 132.6, 126.5, 124.9, 123.1, 111.7, 0.00(TMS). HRMS (ESI-TOF), *m*/*z*: calcd for C_17_H_15_N_4_S_4_Si [M + H]^+^, 430.9943, found, 430.9928. MS (EI, 70eV), *m*/*z* (I, %): 432 ([M + 2]^+^, 1), 431 ([M + 1]^+^, 2), 430 ([M]^+^, 8), 402 (7), 305 (6), 200 (10), 175 (12), 93 (45), 69 (100), 45 (30).

4-Phenylbenzo[1,2-*d*:4,5-*d*′]bis([1,2,3]thiadiazole) (**7f**).

Yellow solid, 125 mg (45%, procedure C), eluent-CH_2_Cl_2_/hexane, 1:2 (*v*/*v*). R_f_ = 0.3 (CH_2_Cl_2_:hexane, 1:1, (*ν*/*ν*)). Mp =203–205 °C. IR ν_max_ (KBr, cm^−1^): 1637, 1492, 1431, 1386, 1277, 1148, 1075, 893, 862, 813, 745, 696, 673, 623, 545, 523. ^1^H NMR (300 MHz, CDCl_3_): δ 9.28 (s, 1H), 7.99 (d, *J* = 6.7, 2H), 7.69–7.59 (m, 3H). ^13^C NMR (100 MHz, CDCl_3_): δ 157.9, 155.6, 140.8, 140.1, 136.9, 130.3, 129.9, 129.7, 129.3, 112.7. HRMS (ESI-TOF), *m*/*z*: calcd for C_12_H_7_N_4_S_2_ [M + H]^+^, 271.0107, found, 271.0109. MS (EI, 70eV), *m*/*z* (I, %): 270 ([M+]^+^, 3), 242 (58), 214 (26), 170 (23), 145 (90), 93 (20), 69 (100), 28 (40).

4-(*p*-Tolyl)benzo[1,2-*d*:4,5-*d*′]bis([1,2,3]thiadiazole) (**7g**).

Green solid, 131 mg (45%, procedure C), eluent-CH_2_Cl_2_/hexane, 1:2 (*v*/*v*). R_f_ = 0.3 (CH_2_Cl_2_:hexane, 1:1, (*ν*/*ν*)). Mp = 229–232 °C. IR ν_max_ (KBr, cm^−1^): 2925, 1639, 1609, 1507, 1427, 1379, 1331, 1317, 1291, 1275, 1192, 1147, 1120, 895, 865, 828, 804, 763, 716, 670, 609, 556, 536, 488. ^1^H NMR (300 MHz, CDCl_3_): δ 9.24 (s, 1H), 7.89 (d, *J* = 7.9, 2H), 7.46 (d, *J* = 7.9, 2H), 2.51 (s, 3H). ^13^C NMR (100 MHz, CDCl_3_): δ 157.8, 155.5, 140.6, 140.4, 139.8, 134.0, 130.0, 129.8, 129.4, 112. 1, 21.3. HRMS (ESI-TOF), *m*/*z*: calcd for C_13_H_8_BrN_4_S_2_ [M + H]^+^, 285.0263, found, 285.0266. MS (EI, 70eV), *m*/*z* (I, %): 284 ([M]^+^, 3), 256 (8), 227 (5), 159 (25), 139 (5), 93 (7), 69 (100), 63 (7), 51 (10), 39 (30), 28 (45), 18 (70).

4-(4-Methoxyphenyl)benzo[1,2-*d*:4,5-*d*′]bis([1,2,3]thiadiazole) (**7h**).

Orange solid, 185 mg (60%, procedure C), eluentCH_2_Cl_2_/hexane, 1:2 (*v*/*v*). R_f_ = 0.2 (CH_2_Cl_2_:hexane, 1:1, (*ν*/*ν*)). Mp = 198–201 °C. IR ν_max_ (KBr, cm^−1^): 3076, 1609, 1509, 1457, 1430, 1383, 1300, 1279, 1262, 1178, 1150, 1116, 1030, 896, 863, 835, 806, 670, 540. ^1^H NMR (300 MHz, CDCl_3_): δ 9.22 (s, 1H), 7.97 (d, *J* = 8.8, 2H), 7.17 (d, *J* = 8.8, 2H), 3.95 (s, 3H). ^13^C NMR (100 MHz, CDCl_3_): δ 161.2, 158.0, 155.6, 140.8, 139.4, 131.2, 129.9, 129.2, 114.8, 112.0, 55.6. HRMS (ESI-TOF), *m*/*z*: calcd for C_13_H_9_N_4_OS_2_ [M + H]^+^, 301.0212, found, 301.0215. MS (EI, 70eV), *m*/*z* (I, %): 302 ([M + 2]^+^, 3), 301 ([M + 1]^+^, 4), 300 ([M]^+^, 30), 272(50), 229 (45), 201 (25), 175 (80), 132 (65), 93 (35), 69 (100), 28 (30).

Methyl 4-(benzo[1,2-*d*:4,5-*d*′]bis([1,2,3]thiadiazole)-4-yl)benzoate (**7i**).

Green solid, 152 mg (45%, procedure C), eluent-CH_2_Cl_2_/hexane, 1:2 (*v*/*v*). R_f_ = 0.1 (CH_2_Cl_2_:hexane, 1:1, (*ν*/*ν*)). Mp = 235–237 °C. IR ν_max_ (KBr, cm^−1^): 2956, 2925, 2854, 1724, 1608, 1463, 1431, 1377, 1277, 1189, 1110, 1084, 1018, 965, 895, 867, 839, 811, 754, 702, 632. ^1^H NMR (300 MHz, CDCl_3_): δ 9.33 (s, 1H), 8.31 (d, *J* = 8.0, 2H), 8.07 (d, *J* = 8.1, 2H), 4.01 (s, 3H). ^13^C NMR (100 MHz, CDCl_3_): δ 166.4, 158.0, 155.5, 141.0, 140.8, 140.1, 131.7, 130.5, 129.8, 128.6, 113.6, 52.5. HRMS (ESI-TOF), *m*/*z*: calcd for C_14_H_9_N_4_O_2_S_2_ [M + H]^+^, 329.0161, found, 329.0151. MS (EI, 70eV), *m*/*z* (I, %): 329 ([M + 1]^+^, 2), 328 ([M]^+^, 8), 300 (100), 256 (10), 227 (12), 213 (30), 203 (65), 144 (45), 93 (10), 69 (80), 59 (8).

4-(Benzo[1,2-*d*:4,5-*d*′]bis([1,2,3]thiadiazole)-4-yl)-*N*,*N*-diphenylaniline (**7j**).

Orange solid, 180 mg (40%, procedure C), eluent-CH_2_Cl_2_/hexane, 1:2 (*v*/*v*). R_f_ = 0.25 (CH_2_Cl_2_:hexane, 1:1, (*ν*/*ν*)). Mp = 213–215 °C. IR ν_max_ (KBr, cm^−1^): 1727, 1590, 1487, 1428, 1321, 1276, 1195, 1125, 1073, 894, 865, 835, 808, 748, 696, 624, 512. ^1^H NMR (300 MHz, CDCl_3_): δ 9.18 (s, 1H), 7.88 (d, *J* = 8.8, 2H), 7.35 (t, *J* = 7.8 Hz, 3H), 7.28–7.12 (m, 9H). ^13^C NMR (100 MHz, CDCl_3_): δ 158.0, 155.3, 150.0, 146.9, 140.8, 139.3, 130.6, 129.8, 129.6, 129.0, 125.7, 124.3, 121.4, 111.5. HRMS (ESI-TOF), *m*/*z*: calcd for C_24_H_15_N_5_S_2_ [M]^+^, 437.0763, found, 437.0757. MS (EI, 70eV), *m*/*z* (I, %): 438 ([M + 1]^+^, 8), 437 ([M]^+^, 55), 409 (6), 381 (4), 312 (12), 168 (3), 69 (15), 18 (100).

## 4. Conclusions

The study of direct palladium-catalyzed (het)arylation reactions of strong electron-withdrawing benzo[1,2-*d*:4,5-*d*′]bis([1,2,3]thiadiazoles showed that this method is useful for the synthesis of mono- and bis-arylated derivatives of this heterocyclic system. Mono- and bis-thienylated benzo-bis-thiadiazoles were selectively obtained by the reaction of 4,8-dibromobenzo[1,2-*d*:4,5-*d*′]bis([1,2,3]thiadiazole) with thiophenes catalyzed by palladium acetate in the presence of potassium pivalate as a base, and no reaction occurred for substituted arenes. The catalytic system, containing Pd(OAc)_2_ and di-*tert*-butyl(methyl)phosphonium tetrafluoroborate salt, proved to be the best for the synthesis of (het)arylated benzo-bis-thiadiazoles from unsubstituted benzo[1,2-*d*:4,5-*d*′]bis([1,2,3]thiadiazole and halogen (bromine or better iodine) (het)arenes. Bis(thienyl)benzo-bis-thiadiazoles were successfully prepared by oxidative hetarylation of benzo[1,2-*d*:4,5-*d*′]bis([1,2,3]thiadiazole with 2-unsubstituted thiophenes, palladium (II) acetate and silver (I) oxide in DMSO.

## Data Availability

Not applicable.

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
