# Peer review of "Palladium-Catalyzed Direct (Het)arylation Reactions of Benzo[1,2-d:4,5-d′]bis([1,2,3]thiadiazole and 4,8-Dibromobenzo[1,2-d:4,5-d′]bis([1,2,3]thiadiazole)"

_molecules, 2023, doi:10.3390/molecules28093977_

Round 1
Reviewer 1 Report
The present work by Rakitin and coworkers describes the heteroarylation of a tricyclic heterocycle. The work is scientifically sound and well performed, even if the level of novelty is not high, and is worth being published on Molecules. I only have a couple of chemical points a few more editorial ones to raise.
Chemical points:
1) In Table 2, entries 10 and 11 appear to differ only by the reaction time (36 h and 24 h respectively). However the bis-arylated product is observed after a 24 h reaction time, but not after 36 h. This is quite surprising. Please comment.
2) The NMR spectra reported in the SI are clean and clear, but only the spectra of new compounds are reported. For known compounds, it is only said in the text that the obtained spectra are consistent with literature data. Most journals nowadays require the spectra even of known compounds to be reported to allow the reviewers to check the purity. I don't know which is the policy of Molecules in this respect. If it is that they should be present, they must be added.
Editorial points:
1) All tables lack a footnote. A table footnote should be added specifying all experimental conditions, specifically the reagents, catalyst, ligand and solvent amounts. Only the ratio between the two reagents is reported. The reader should see this information without having to go to the experimental part.
2) Page 2, line 78: Pd2(dba)3 in place of Pd2dba3
3) Caption of Table 1: 3a-e, but only a-d are shown and employed.
4) Paragraph 2.2. In the full paragraph, compound 4a should be 7a
5) Page 5, lines 169-170. For the iodo derivative, entry 15 should be compared with entry 11 and I would not say that the iodo derivative gave worse results.
6) Page 7. Insert "of" between "instead" and "palladium"
7) In the conclusions, the authors honestly discuss the pros and the cons of the reported synthetic strategies with respect to those known from the literature. However, I noted that the palladium loading used is quite high, up to 27 mol%. This is likely quite higher that that required when using Suzuki or Stille coupling and this aspect should be mentioned.
8) Page 10, paragraph 3.4. A sentence is reported sounding ".. were added to an air free flask which was then purged in dry xylene (8 mL) was degassed by argon and heated at 130 °C in a sealed vial." This sentence makes little sense and should be rephrased. The same applying even to the following recipes.
See comments above.
Author Response
The authors are grateful to the reviewer for a kind and highly professional review.
Reviewer 1:
In Table 2, entries 10 and 11 appear to differ only by the reaction time (36 h and 24 h respectively). However the bis-arylated product is observed after a 24 h reaction time, but not after 36 h. This is quite surprising. Please comment.
Authors:
Entry 11 in Table 2 has been removed.
Reviewer 1:
The NMR spectra reported in the SI are clean and clear, but only the spectra of new compounds are reported. For known compounds, it is only said in the text that the obtained spectra are consistent with literature data. Most journals nowadays require the spectra even of known compounds to be reported to allow the reviewers to check the purity. I don't know which is the policy of Molecules in this respect. If it is that they should be present, they must be added.
Authors:
NMR spectra of the known compounds are added to the Experimental part of the paper and to the SI.
Reviewer 1:
All tables lack a footnote. A table footnote should be added specifying all experimental conditions, specifically the reagents, catalyst, ligand and solvent amounts. Only the ratio between the two reagents is reported. The reader should see this information without having to go to the experimental part.
Authors:
A table footnotes are added to the Tables 1-4.
Reviewer 1:
2) Page 2, line 78: Pd2(dba)3 in place of Pd2dba3
Authors:
Corrected as suggested by the Reviewer.
Reviewer 1:
3) Caption of Table 1: 3a-e, but only a-d are shown and employed.
Authors:
Corrected as suggested by the Reviewer.
Reviewer 1:
4) Paragraph 2.2. In the full paragraph, compound 4a should be 7a
Authors:
Corrected as suggested by the Reviewer.
Reviewer 1:
5) Page 5, lines 169-170. For the iodo derivative, entry 15 should be compared with entry 11 and I would not say that the iodo derivative gave worse results.
Authors:
The sentence is removed from the text of the paper.
Reviewer 1:
6) Page 7. Insert "of" between "instead" and "palladium"
Authors:
Corrected as suggested by the Reviewer.
Reviewer 1:
7) In the conclusions, the authors honestly discuss the pros and the cons of the reported synthetic strategies with respect to those known from the literature. However, I noted that the palladium loading used is quite high, up to 27 mol%. This is likely quite higher that that required when using Suzuki or Stille coupling and this aspect should be mentioned.
Authors:
Unfortunately, the paper presented slightly outdated data; we found that reducing the amount of the palladium catalyst to 15 mol%, which is optimal for the reactions under study, does not reduce the yields of the final products. In the experimental part of the paper, the amount of palladium catalyst was replaced by 15 mol%.
Reviewer 1:
8) Page 10, paragraph 3.4. A sentence is reported sounding ".. were added to an air free flask which was then purged in dry xylene (8 mL) was degassed by argon and heated at 130 °C in a sealed vial." This sentence makes little sense and should be rephrased. The same applying even to the following recipes.
Authors:
This sentence is rephrased in all procedures 3.3-3.8.
Reviewer 2 Report
Rakitin and co-workers studied the Pd-catalyzed mono- and bis- heteroarylation of the benzo[1,2-d:4,5-d']bis([1,2,3]thiadiazole) and of the 4,8-dibromobenzo[1,2-d:4,5-d']bis([1,2,3]thiadiazole). The reaction proceeds only in the presence of thiophene derivatives, and the obtainment of the mono- or bis-substituted product is driven essentially by the amount of the thiophene. Under different reaction conditions, generally modest yields of the desired products are obtained.
In the present state, the manuscript is not suitable for publication on Molecules, therefore the following major revision and considerations have to be taken into account.
1) A very careful language review is strongly recommended.
2) Abstract: please substitute benzo[1,2-d:4,5-d']bis([1,2,3]thiadiazole with benzo[1,2-d:4,5-d']bis([1,2,3]thiadiazole). The final parenthesis is missing.
3) Page 2, lines 93-94: since the authors wrote that the double oxidative CH functionalization was discovered, at least the year and possibly the research group should be indicated in the text.
4) Page 3, Scheme 1: general conditions for each path should be added on the arrow. This would greatly facilitate the reading of the scheme.
5) Page 4, line 122-124. Please remove “the reaction conditions:”.
6) Page 5: on Table 2 is reported product 7a. However, in the text the authors indicating the mono-arylated product refer to 4a, which was the product deriving from the mono-arylation of the 4,8-dibromobenzo[1,2-d:4,5-d']bis([1,2,3]thiadiazole). Please, correct the text.
7) Page 5: every different molecule should be labelled differently. Therefore, 6a cannot refer to 2-iodothiophene, 2-bromothiphene and 2-chlorothiophene derivatives. Please, label different molecules differently. The same should also be done for the thiophene derivatives of Table 3 and, more generally, throughout the text.
8) Scheme 3: the scheme is not clear. In particular, in the scheme should be immediately visible which reactions were performed by the authors and which ones are taken by the literature. Please edit the scheme.
9) Experimental Section, line 135: please, indicate the “appropriate drying agents” utilized for the purification of the solvents.
10) Experimental Section, point 3.5, line 305: the benzo[1,2-d:4,5-d']bis([1,2,3]thiadiazole) has been labelled with 1. Please, substitute 2 with 1 (it should be also bolded). The same should be also done for the point 3.6 (line 316)
11) 4-Bromo-8-(thiophen-2-yl)benzo[1,2-d:4,5-d']bis([1,2,3]thiadiazole) (4b), line 352: Please, double check that the cited literature for the mp is correct.
12) 4,8-Bis(5-(2-ethylhexyl)thiophen-2-yl)benzo[1,2-d:4,5-d']bis([1,2,3]thiadiazole) (5a), line 366: the correct Rf eluent is missing.
13) Even for already reported products at least the 1H-NMR characterization should be reported in the experimental section, together with the already present sentence that claims it corresponds to literature data.
14) For all the isolated products in the present work (both news and already reported), copies of 1H-NMR and 13C-NMR spectra have to be reported in the Supporting Information.
15) The authors both in the title and in the text refer to (het)arylation reactions, although only with thiophene derivatives the reaction succeeded, since with toluene and xylene it fails. What happens if other heteroarenes (e.g., furan or thiazole) are tested?
A very careful language review is strongly recommended.
Author Response
The authors are grateful to the reviewer for highly professional review.
Reviewer 2:
1) A very careful language review is strongly recommended.
Authors:
We have carefully checked the manuscript and corrected the mistakes.
Reviewer 2:
2) Abstract: please substitute benzo[1,2-d:4,5-d']bis([1,2,3]thiadiazole with benzo[1,2-d:4,5-d']bis([1,2,3]thiadiazole). The final parenthesis is missing.
Authors:
Corrected as suggested by the Reviewer.
Reviewer 2:
3) Page 2, lines 93-94: since the authors wrote that the double oxidative CH functionalization was discovered, at least the year and possibly the research group should be indicated in the text.
Authors:
The necessary information is added to the text of the paper.
Reviewer 2:
4) Page 3, Scheme 1: general conditions for each path should be added on the arrow. This would greatly facilitate the reading of the scheme.
Authors:
The general conditions for every reaction are added to the Scheme 1.
Reviewer 2:
5) Page 4, line 122-124. Please remove “the reaction conditions:”.
Authors:
Corrected as suggested by the Reviewer.
Reviewer 2:
6) Page 5: on Table 2 is reported product 7a. However, in the text the authors indicating the mono-arylated product refer to 4a, which was the product deriving from the mono-arylation of the 4,8-dibromobenzo[1,2-d:4,5-d']bis([1,2,3]thiadiazole). Please, correct the text.
Authors:
Corrected as suggested by the Reviewer.
Reviewer 2:
7) Page 5: every different molecule should be labelled differently. Therefore, 6a cannot refer to 2-iodothiophene, 2-bromothiphene and 2-chlorothiophene derivatives. Please, label different molecules differently. The same should also be done for the thiophene derivatives of Table 3 and, more generally, throughout the text.
Authors:
Each compound received its own number, for example, 6a(Br).
Reviewer 2:
8) Scheme 3: the scheme is not clear. In particular, in the scheme should be immediately visible which reactions were performed by the authors and which ones are taken by the literature. Please edit the scheme.
Authors:
Reactions that have been performed previously are marked with a reference number.
Reviewer 2:
9) Experimental Section, line 135: please, indicate the “appropriate drying agents” utilized for the purification of the solvents.
Authors:
The necessary information is added.
Reviewer 2:
10) Experimental Section, point 3.5, line 305: the benzo[1,2-d:4,5-d']bis([1,2,3]thiadiazole) has been labelled with 1. Please, substitute 2 with 1 (it should be also bolded). The same should be also done for the point 3.6 (line 316)
Authors:
Corrected as suggested by the Reviewer.
Reviewer 2:
11) 4-Bromo-8-(thiophen-2-yl)benzo[1,2-d:4,5-d']bis([1,2,3]thiadiazole) (4b), line 352: Please, double check that the cited literature for the mp is correct.
Authors:
The number of the reference is corrected
Reviewer 2:
4,8-Bis(5-(2-ethylhexyl)thiophen-2-yl)benzo[1,2-d:4,5-d']bis([1,2,3]thiadiazole) (5a), line 366: the correct Rf eluent is missing.
Authors:
The correct Rf eluent is added.
Reviewer 2:
13) Even for already reported products at least the 1H-NMR characterization should be reported in the experimental section, together with the already present sentence that claims it corresponds to literature data.
Authors:
NMR spectra of the known compounds are added to the Experimental part of the paper and to the SI.
Reviewer 2:
14) For all the isolated products in the present work (both news and already reported), copies of 1H-NMR and 13C-NMR spectra have to be reported in the Supporting Information.
Authors:
NMR spectra of the known compounds are added to the Experimental part of the paper and to the SI.
Reviewer 2:
15) The authors both in the title and in the text refer to (het)arylation reactions, although only with thiophene derivatives the reaction succeeded, since with toluene and xylene it fails. What happens if other heteroarenes (e.g., furan or thiazole) are tested?
Authors:
Unfortunately, we have not studied reactions with furan or thiazole derivatives. We have investigated reactions with various thiophene derivatives, since they are the most interesting for the design of components for organic light emitting diodes and solar cells.
Round 2
Reviewer 2 Report
The revised paper is suitable for publication on Molecules